# Risk Factors for *Salmonella* Detection in Commercial Layer Flocks in Spain

**DOI:** 10.3390/ani13203181

**Published:** 2023-10-12

**Authors:** Clara Samper-Cativiela, Maria Esther Prieto, Soledad Collado, Cristina De Frutos, Adam J. Branscum, Jose Luis Saez, Julio Alvarez

**Affiliations:** 1VISAVET Health Surveillance Centre, Universidad Complutense, 28040 Madrid, Spain; 2Department of Animal Health, Faculty of Veterinary Medicine, Universidad Complutense, 28040 Madrid, Spain; 3Ministerio de Agricultura, Pesca y Alimentación, 28010 Madrid, Spainscollado@mapa.es (S.C.);; 4Biostatistics Program, School of Biological and Population Health Sciences, Oregon State University, Corvallis, OR 97331, USA; adam@oregonstate.edu

**Keywords:** *Salmonella*, poultry, laying hen, national control program, epidemiology, risk factor, Bayesian hierarchical model

## Abstract

**Simple Summary:**

Foodborne salmonellosis remains one of the top zoonotic diseases affecting public health worldwide, and its incidence has remained stable in the last years in the European Union (EU) triggering questions on the usefulness of currently available measures to prevent its occurrence. A main focus of *Salmonella* national control programs is monitoring the presence of the bacteria in animal reservoirs, especially in poultry, and for this reason, thousands of samples are collected every year in poultry farms in EU countries, but the importance of certain factors in the probability of detecting *Salmonella* remains poorly understood. A thorough analysis conducted on data collected in all laying hen flocks sampled in Spain in 2015–2020 revealed that even though the presence of *Salmonella* was rare (<3.5% of positive sampling events), when samples were collected in certain months (fall–winter) and housing systems (caged flocks) and by competent authorities (as opposed to food business operators), the probability of detecting *Salmonella* increased significantly. These results demonstrate that the sensitivity of the sampling strategy may be influenced by how and when samples are collected and that certain flocks may be at an increased risk of infection.

**Abstract:**

Trends in *Salmonella* human infections are assumed to be related to the distribution of the pathogen in the animal reservoir/food products, and cases in humans are most often linked to poultry and poultry products (eggs, meat). Therefore, ongoing *Salmonella* national control programs (NCPs) in European Union Member States have the objective of monitoring and reducing its prevalence in commercial poultry flocks. Results from NCPs have shown certain factors (housing systems, season of sampling and if sampling is conducted by food business operators (FBOps) or competent authorities (CAs), among others) can influence detection rates, but associations are often not consistent. Here, we analyzed data from the Spanish NCP on 7216 laying hen flocks subjected to 36,193 sampling events over a six-year period to characterize its performance and identify variables influencing detection rates. Overall, 1205 sampling events were positive for *Salmonella* spp. (any serovar) and 132 for *S*. Enteritidis-*S*. Typhimurium/monophasic. Bayesian multivariable models adjusting for multiple covariates concluded that sampling events later in the year, in caged flocks with older animals and conducted by CAs had increased odds of positivity for *Salmonella* spp., revealing aspects linked with a differential estimation of *Salmonella* levels in laying hen flocks.

## 1. Introduction

Despite significant efforts to prevent and control non-typhoidal *Salmonella enterica* subsp. *enterica* (NTS) infection, it remains one of the top causes of foodborne diseases in low-, medium- and high-income countries, with over 75,000 cases reported annually in Europe between 2016 and 2019 [1], although this likely represents a severe underestimation [2]. Even though the number of confirmed salmonellosis cases in the European Union (EU) decreased between 2008 and 2012, case counts have stabilized in recent years (2016–2020) [1].

Transmission of non-typhoidal salmonellosis is usually due to ingestion of contaminated food, with pork and, especially in the case of outbreaks, poultry products (in particular eggs and egg-based products) often identified as the main vehicles of infection [3,4,5]. Even though multiple NTS serovars can cause human infection, most cases are due to a relatively low number of them, with *S*. Enteritidis and *S*. Typhimurium being identified as the two top disease-causing serovars in Europe and worldwide [1]. Because of the perceived importance of poultry as a source of NTS, numerous countries have implemented control programs to monitor the presence of *Salmonella* in production flocks and prevent the transmission of NTS through the food chain. In Europe, the establishment of national control programs (NCPs) for *Salmonella* serovars of “public health significance” in all relevant stages of production, processing and distribution in poultry is stipulated in Regulation (EC) 2160/2003. In the case of laying hens of *Gallus gallus*, responsible for an estimated 5.7–22.2% of all human salmonellosis cases in the EU in 2016 [6], the goal of NCPs is to maintain the percentage of positive flocks of adult laying hens at ≤2% or to ensure an annual percent reduction in prevalence of at least 10% (if prevalence in the previous year was below 10%), ≥20% (if prevalence in the previous year was between 10% and 20%), ≥30% (if prevalence in the previous year was between 20% and 40%) or ≥40% (if prevalence in the previous year was at or above 40%) according to Commission Regulation 517/2011. Commission Regulation 517/2011 also lists the serovars with public health significance in the case of laying hen flocks (*S*. Enteritidis and *S*. Typhimurium including its monophasic variant with the antigenic formula 1,4,[5],12:i:-, referred from here on as target serovars).

Control of *Salmonella* infection in laying hen flocks is complicated by its complex dynamics in infected flocks, which are influenced by multiple factors including management and host factors and the multiple possible sources of infection, such as entry of new (infected) animals or contaminated eggs, domestic and wild reservoirs, insects and contaminated equipment, feed or water [7]. The use of antimicrobials to control *Salmonella* infection in poultry is not allowed in the EU, and therefore, adequate biosecurity measures and vaccination are the main tools to prevent the establishment of NTS infections in layer flocks [8]. Once NTS is present in a farm, infected animals may act as asymptomatic carriers, perpetuating the infection in the flock and further contaminating the environment and the food products produced [9].

According to Commission Regulation 517/2011, detection of the presence of NTS in layer flocks by NCPs is based on periodical sampling events conducted by the food business operator (FBOp) and the competent authorities (CAs) at different frequencies (see Material and Methods). In addition, the CA must also carry out additional sampling in other situations: in flocks present in a farm where a flock has tested positive for a target serovar or in new flocks housed in the same building where a previously positive flock was present; in the case of suspicion due to, for example, foodborne outbreak investigation; and in any other situation where the CA considers it appropriate. Sampling protocols vary depending on the housing system (see Materials and Methods). The different sampling strategies, along with factors such as the housing system (e.g., caged vs. cage-free flocks), flock size and flock age, have been shown to influence the probability of positive test results [6,10,11], which could be attributable to both an increased prevalence of infection in certain flocks/periods and variability in the performance of different surveillance protocols [12]. The sampler (FBOp or CA) has also been shown to influence results from the EU *Salmonella* NCPs in poultry housing systems in which separate reporting is mandatory (broilers and fattening and breeding turkeys), with samples collected by CAs having a significantly higher likelihood of a test positive result, prompting the need to conduct further investigations to define the reasons behind such discrepancies [1]. However, depending on the country, separate data for FBOps and CAs is also available for other poultry categories, such as laying hens in Spain. Currently, it is unclear whether this is also the case in laying hens since there is no requirement for separate reporting of CA/FBOp data, an issue worth investigating as recently recommended by EFSA and ECDC [1]. 

In this context, we conducted an analysis of the data from NCPs performed in laying hen flocks in Spain from 2015 to 2020 to (1) characterize the sampled population and the frequency of detection of *Salmonella* spp. and of target serovars and (2) identify factors associated with increased probability of *Salmonella* detection related to the flock and the sampling strategy while accounting for other potentially influencing factors. 

## 2. Materials and Methods

### 2.1. Study Population

This study was performed using data from the NCP for *Salmonella* in Spain carried out between 2015 and 2020 according to the European and national legislation (Commission Regulation 517/2011, Spanish Royal Decrees 328/2003 and 637/2021 and the annually approved NCP for *Salmonella*). Briefly, these NCPs are based on periodical sampling events conducted by the FBOp (according to the EU guidelines, adult flocks of laying hens are sampled at least every 15 weeks, with the first sampling in the production stage taking place at a flock age of 24 ± 2 weeks, but the NCP for Salmonella in Spain starts with 1-day-old birds, with sampling events already at the rearing stage) or the CA (that will sample at least one adult flock per farm comprising at least 1000 birds, preferably at the end of the production phase, in addition to other scenarios—see below “reasons for sampling”). Samples collected may include pooled feces (in birds housed in cages), boot swabs (in barn or free-range houses) and dust samples (collected either directly or through the use of fabric swabs). Samples are then transported to authorized laboratories where they will be processed for the isolation of *Salmonella* according to ISO6579-1:2003 and ISO6579-1:2017, the method recommended by the European Union Reference Laboratory for *Salmonella* in fecal and environmental samples, as described in Commission Regulation 517/2011. If a positive result is found (i.e., bacterial growth is obtained), at least the serotyping for target serovars must be carried out to confirm/discard the presence of *S*. Enteritidis and *S*. Typhimurium (including its monophasic variant). 

The database analyzed here contained information on all samples collected from laying hen flocks for the detection of *Salmonella* spp. using bacteriology and included samples collected by CA and FBOp. The database included information on several variables at the different hierarchical levels: Farm level: location of the farm (municipality);Flock level: size of the flock (number of birds) and housing type;Sampling level: date of sampling (day), reason for sampling (see below), sampler (CA or FBOp), number of samples analyzed independently and age of the flock at sampling;Sample level: sample type (i.e., specimen) and laboratory results from the sample (isolation of *Salmonella* and, when positive, if a target serotype—Enteritidis or Typhimurium—was identified).

### 2.2. Data Preparation

Samples of a given type (e.g., boot swabs, feces and dust) collected on the same day were grouped in sampling events, which represented the unit of analysis. The hierarchical structure of the data therefore had three levels: farms, flocks (with farms typically housing multiple flocks over the study period) and sampling events (with flocks typically being sampled multiple times over the study period). In this study, flocks were defined as a group of birds entering the same day in a farm and housed in the same barn so that they share the same air space. Prior to data analysis, variables collected at the different levels were categorized/recategorized as follows:

*Farm-level variables*: Since very few farms were sampled per municipality or province, location was considered at the autonomous region (from here on, “region”) level. Regions with less than 50 farms were collapsed into a single category (“Region 8”). 

*Flock-level variables*: Size of flock was categorized into quartiles by using the empirical 25th, 50th and 75th percentiles as cutoffs. Housing type was analyzed using four categories (birds housed in cages, floor, free-ranging or organic systems) and as a dichotomous variable (cage systems vs. other systems).

*Sampling-level variable*: Date of sampling was categorized by year as well as bimester (Jan–Feb, Mar–Apr, May–Jun, Jul–Aug, Sep–Oct and Nov–Dec). There were 13 options for reason for sampling, many of which were associated with a previous isolation of *Salmonella* (confirmation of a positive result, sampling events conducted in laying flocks (of age 24 ± 2 weeks) housed in premises in which the previous flock was positive, control of all other flocks after a positive flock has been detected in a farm, epidemiological investigations due to connections with a positive flock/foodborne outbreak) or other reasons (e.g., tests of antimicrobial usage). Given that the objective of the study was to identify variables associated with *Salmonella* detection in routine sampling events conducted in the frame of the NCP (i.e., in the absence of any suspicion) and that all the previously mentioned categories accounted for a very small proportion of all observations in the initial database (<2.5%), they were excluded from further analysis leaving only environmental (i.e., sampling conducted to verify the efficacy of cleaning and disinfection practices conducted by the FBOp) and periodical sampling as possible values. In the analyses, these were combined with the sampler (CA/FBOp) variable so that three categories remained: FBOp_periodical, FBOp_environmental and CA_periodical. Environmental sampling events conducted by the CA were excluded since these were typically performed after cleaning and disinfection of *Salmonella*-positive flocks, while environmental samplings conducted by FBOp were maintained since they were not typically linked with positive flocks but conducted usually after cleaning and disinfection once a barn had been emptied before the new flock arrived irrespective of the status of the emptied flock. Thus, this category was maintained separately in the analysis to detect possible variations in detection rates in this context.

For age, due to the detection of abnormally low and high recorded values and the lack of variation in the age of a single flock at different FBOp sampling events in some cases in the database, a quality check was conducted. This consisted of comparing the expected age of the birds (based on the difference between the sampling date and the date of entry of the animals assuming they entered at 17 weeks of age) and the recorded age at the time of first CA sampling or, for flocks in which no CA sampling had been conducted, in the first FBOp sampling. Only observations for which the expected and the recorded date differed by no more than 6 weeks (an arbitrary threshold selected to exclude only records where a significant deviation was suspected) were maintained in the database, and the recorded age in the initial CA/FBOp sampling was used as a reference to calculate the age of the flock at subsequent sampling events by adding the difference (days) between the first sampling and each subsequent sampling. 

Sample type, once sampling events due to non-routine reasons were excluded, could be one of boot swabs, feces, dust and fabric swabs, and each sampling could involve the collection of one or more samples. Finally, the laboratory results were categorized into two dichotomous variables, each based on whether one or more of the samples collected in the frame of a given sampling yielded a positive result: isolation of (any serovar of) *Salmonella* (yes/no) and isolation of a target serovar, i.e., Typhimurium—including its monophasic variant—or Enteritidis (yes/no).

Variables available for the study and their nature are summarized in Appendix A.

### 2.3. Data Analysis

Two models with different response variables were considered for data analysis: one based on isolation of *Salmonella* (any serovar) and the other based on isolation of target serovar (only Enteritidis or Typhimurium including its monophasic variant). In order to assess the relationship between the available covariates at different hierarchical levels and the response variables of interest, a Bayesian mixed-effects logistic regression model was fitted that accounted for the hierarchical structure of the data and within-farm and within-flock correlations. The laboratory result (positive/negative) obtained in sampling “*i*” conducted in flock “*j*” housed in farm “*k*” was assumed to follow a Bernoulli distribution:$$y_{i,j,k}\sim Bernoulli\left(p_{i,j,k}\right),$$
with $p_{i,j,k}$ being the probability of obtaining a positive laboratory result (isolation of *Salmonella* spp. or isolation of a target serovar). This probability was modeled as a function of the variables available at the sampling level and, in order to account for the dependence between flocks housed in the same farm and of sampling events conducted in the same flock, we included random effects for farm and flock (nested in farm):$$logit\left(p_{i,j,k}\right)=U_{j,k}+X_{i,j,k}\beta$$
where $X_{i,j,k}$ denotes the vector of covariates (e.g., date of sampling) for sampling *i* in flock *j* from farm *k,* with β being the corresponding vector of regression coefficients. The random effect for flock-level ($U_{j,k}$) was modeled as
$$U_{j,k}\sim Normal\left({\mu \_flock}_{j,k},\sigma_{flock}\right),$$
with mean that depends on a vector of flock-level covariates ($Z_{j,k}$, e.g., housing type), with corresponding coefficient vector *γ* and a farm-specific term *α_k_*: $${\mu \_flock}_{j,k}=\alpha_{k}+Z_{j,k}\gamma$$

Finally, the farm-level random effect was modeled as a function of the only covariate available at that level, namely the farm location ($V_{k}$):$$\alpha_{k}\sim Normal\left({\mu \_farm}_{k},\sigma_{farm}\right),$$
where ${\mu \_farm}_{k}=\delta_{0}+\delta_{1}V_{k}$.

The variable number of analyses was excluded from the multivariable analysis due to its correlation with the sampler (93.6% of samples conducted by CA involving more than one sample vs. 1.1% in the case of those conducted by FBOp).

Independent diffuse Gaussian prior distributions with mean 0 and variance 10 restricted to the (−10, 10) range were assumed for the regression coefficients *β*, *γ* and *δ*, while uniform distributions U(0.01, 10) were used for the variance components *σ*_flock_ and *σ*_farm_. The association between a given covariate and the probability of a positive test result for *Salmonella* was evaluated using the posterior medians and 95% posterior probability intervals (PPIs) for odds ratios. Variables were included in the final model depending on their association (based on the 95% PPI) and the deviance information criteria (DIC), such that the model with the lowest DIC was preferred [13]. The predictive ability of the final model was assessed via a receiver operating curve (ROC) using the model-based predicted probability of the model as the continuous response and the observed binary outcome as the classification variable.

Models were fitted in OpenBUGS 3.2.3 [14] through the R2OpenBUGS package [15] in R 4.0.2 [16]. Markov Chain Monte Carlo (MCMC) runs were performed using three chains with different randomly selected initial values, and chains were thinned to avoid autocorrelation by selecting one in every 10 consecutive samples. Convergence was assessed visually by examining the mixing of the chains and more formally using the Gelman–Rubin statistic [17,18]. Models were run for 2500 iterations after discarding the first 500 burn-in samples. Model checking was performed through posterior predictive simulations in which the observed data were compared with the replicated datasets generated from the posterior predictive distribution of the final model across different levels of the predictors considered.

## 3. Results

### 3.1. Descriptive Results

Upon exclusion of non-routine sampling events and flocks in which the age recorded was not considered reliable, the database analyzed contained initially 36,193 sampling events conducted between January 2015 and December 2020 in 7216 flocks housed in 1153 farms. Cataluña, Castilla y Leon, Andalucia and Castilla la Mancha were the autonomous regions with the highest number of sampling events (55.9% of all sampling events), flocks (53.0% of all flocks) and farms (53.3% of all farms), although the relative contribution of each region depended on the level of the hierarchy in the database considered (Figure 1). 

Farms included in the study housed on average 6.3 flocks over the study period (median = 4, IQR = 2–8), and flocks were sampled on average five times while present in the farm (median = 5, IQR = 3–7). The average size of the flocks was 26,316 birds (median = 10,000, IQR = 2500–32,070), and these were most commonly in the cage system (3833/7216, 53.1%%) followed by floor (20.3%), free-ranging (16.5%) and organic flocks (10.1%) (Table 1). However, these proportions were not constant over time, with organic and free-ranging systems becoming more common in the last years of the study period (Figure 2). Caged flocks were much larger than other categories (median size = 29,949 birds vs. 5100 for floor system flocks, 5253 for free-ranging and 1700 for organic).

Most of the sampling events were performed by the FBOp (32,794/36,193, 90.6%) and were most commonly periodical sampling (86.8% of all FBOp sampling events), with the remaining being environmental sampling events. The remaining 3399 sampling events were conducted by the CA and belonged to the periodical sampling category (Table 1). Sampling events conducted by the FBOp were generally based on the analysis of one pool made up of two individual samples (99.6% and 94.4% of all FBOp periodical and environmental sampling events, respectively), with the remaining sampling events considering the result of between two and nine independent bacteriological analyses. In contrast, sampling events performed by the CA typically (93.6% of all CA sampling events) considered the results of two independent bacteriological analyses (of a pool made up of two samples as in those conducted by the FBOp plus an additional sample analyzed separately). The annual number of sampling events conducted ranged between 5112 in 2015 and 7158 in 2020, with a clear increasing trend over the study period (Table 1, Appendix A). In each year, sampling events were most often conducted in the last two bimesters (between September and December). The average age of the sampled flocks was 62.8 weeks (median = 61, IQR = 39–83). Finally, samples collected consisted most often of fecal samples (used in 20,065 sampling events, 55.4%) followed by boot swabs (11,720, 32.4%), both of which were used in the vast majority of the periodical sampling events conducted by both the FBOp and the CA (all but 96 of the 31,880 periodical sampling events included in the database) (Table 1). Dust was the sample of choice in 3542 sampling events (all but 90 classified as environmental), and fabric swab samples were used in 866 sampling events, of which 860 were environmental (Table 1).

### 3.2. Salmonella spp. Detection

*Salmonella* spp. was detected in 1205 of the 36,193 sampling events analyzed throughout the whole study period (3.33% positivity). Detection rates over time depending on the sampler are shown in Appendix A. Among these cases, *S*. Enteritidis or Typhimurium (including its monophasic variant) were identified in 132 sampling events (Enteritidis in 81 sampling events, Typhimurium in 37 and the monophasic variant in 14), representing 0.36% of all sampling events and 11.0% of the *Salmonella* positive cases. Among isolates belonging to non-target serovars, in 382 sampling events (31.9%), no further serotyping was carried out once the presence of target serovars was discarded, and for those that were fully serotyped, the most frequent types were Infantis (retrieved in 136 sampling events, 16.5% of the 823 sampling events in which the *Salmonella* recovered were fully serotyped), Ohio (110 sampling events, 13.4%) and Corvallis (50 sampling events, 6.1%) (Appendix A). More than one serovar was retrieved in 20 sampling events. 

Out of the 1153 farms and 7216 flocks sampled at least once during the study period, 437 farms (37.9%) and 838 flocks (11.6%) were positive at least once. Among the locations with positive sampling events for target serovars, these numbers dropped to 108 farms (9.4%) and 123 flocks (1.7%). At the farm level, the percentage of positive farms varied widely depending on the region, with some regions having 40–60% of their farms testing positive for *Salmonella* spp. at least once while in others, the proportion of positive farms remained below 25%. When only sampling events positive to target serovars are considered, the proportion of positive farms per region dropped to 3.1–18.7% but was highly correlated with the proportion of positive samples to *Salmonella* (any serovar) (Pearson’s rho = 0.83) (Appendix A). When considering the positivity at the flock level, similar regions had higher/lower proportions of positive flocks with again a high correlation between results for *Salmonella* and for target serovars only (Pearson’s rho = 0.86) (Appendix A). 

A clear trend in positive sampling events as a function of flock size was not observed, although lower proportions of positive sampling events for *Salmonella* spp. were found in smaller flocks (9.4% in flocks with <10,000 birds compared to 16.3% in flocks with 10,000–32,055 birds and 11.4% in those with >32,055 birds), while the proportion of positive sampling events for target serovars was consistently around 2.0% for flocks with <32,055 birds and around 0.9% for those above that size (Table 1). Regarding the **housing system**, a larger proportion of positive sampling events for *Salmonella* spp. was found in flocks housed in caged systems (586/3833, 15.3%). For target serovars, the highest proportion of positive sampling events was observed for free-ranging flocks (29/1188, 2.4%) (Table 1). 

Finally, at the sampling level, differences in the proportions of positive sampling events both for *Salmonella* spp. and only for specific serovars were observed for multiple variables. In terms of the **sampler**, sampling events conducted by the CA were more often positive than those conducted by FBOps when considering periodical sampling events (12.7% vs. 2.6% for *Salmonella*, 3.0% vs. 0.08% for target serovars), while FBOp environmental sampling events yielded the lowest proportion of positive results for *Salmonella* spp. (1.0%) but not for target serovars (0.14%). Regarding the **sampling date**, the proportion of positive sampling events every year ranged between 2.9 and 3.6% for all *Salmonella* and between 0.16 and 0.44% for target serovars, with more positive sampling events detected in both cases in the last two bimesters (4.3–4.7% and 0.49–0.62 for *Salmonella* spp. and target serovars) than in the other four (<3% and <0.33%) (Table 1). Concerning the **age of the flock**, a higher prevalence of sampling events tested positive in older (>83 weeks) birds (4.9% positive for *Salmonella* spp. and 0.70% for target serovars) than in younger birds (Table 1). Similarly, and in terms of the **number of analyses**, sampling events based on more than one independent bacteriological analysis were more often positive for all *Salmonella* serovars (11.3%) and for target serovars (2.59%) than those involving a single laboratory analysis (2.5% and 0.12%, respectively, for *Salmonella* spp. and only target serovars). Finally, for the **sample type**, sampling events based on feces were more commonly positive (4.5%) for all serovars compared to those based on boot swabs (2.1%), dust (1.4%) or fabric swabs (1.2%), but differences were reduced when considering target serovars (0.2 to 0.50% positive rate regardless of the sample type) (Table 1).

### 3.3. Model Results

For the response variable of testing positive for any type of *Salmonella* spp. (yes/no), the final Bayesian mixed-effects logistic regression model included all variables considered except the size of the flock and the number of analyses (farm level: region; flock level: housing type as a dichotomous variable; sampling level: sampler reason for sampling, year, bimester, age of the flock at the sampling, sample type). Regression coefficients for the flock-level variables (flock size in quartiles and housing type) could not be estimated when the model included both, due in part to multicollinearity, and housing type was preferred and thus kept in the model. Furthermore, housing type was included in the model as a dichotomous variable (caged vs. other systems) due to the improved DIC over the model containing all four categories and because similar values were obtained for the coefficients of the organic, floor and free-ranging categories when these were considered separately. Finally, when including sample type in the model the dust/fabric swabs categories were merged since both had similar percentages of positivity (Table 1) and were used in environmental sampling events.

**Farm location (region)** was associated with an increased likelihood of finding positive results in sampling events (Table 2). Specifically, compared to the reference region (Region 0) three regions (Region 4, Region 6 and Region 9) had odds ratios indicating a lower risk (upper limit of the 95% PPI below 0.65), while the 95% PPI of the coefficients associated with Regions 2, 3, 7 and 8 barely included 1 (Table 2). In contrast, Region 5 to some extent and particularly Region 1 had an increased risk compared with the reference category, with part or all of their 95% PPI above 1, respectively (Table 1). This was reflected in the estimates of the farm-level random effect α, which was largely influenced by the region where the farm was located (Appendix A). At the flock level and regarding **housing type**, odds ratios greater than 1 were found in caged flocks (median OR = 2.07, 95% PPI 1.52–2.85) compared with the reference category (organic/free-ranging/floor systems) (Table 2), with flock-level random effects influenced by the region where the flock was located though with higher variability compared with the farm-level random effect (Appendix A).

Finally, at the sampling level, a strong effect of the variables **sampler, bimester and age of the flock** was observed, so that higher odds of obtaining a positive result in sampling events conducted by CAs (OR = 6.64, 95% PPI 5.63–7.75), between the months of September and December (95% PPI 1.36–2.12 for Sep–Oct and 0.99–1.56 for Nov–Dec) and in flocks with older (>83 weeks) birds (OR = 1.63, 95% PPI 1.35–2.00) were observed compared with the reference categories for each variable, while lower odds were observed for sampling events in 2015 compared with 2020 (Table 2, Figure 3). The model had a good predictive ability as demonstrated by the ROC curve (AUC = 94.2, 95%CI 93.7–94.8) and the precision–recall curve (Appendix A) 

Model convergence was achieved as demonstrated in part by Gelman–Rubin statistics that were below 1.01 (Table 2) and two farm-level random effects. The model fitted the data well as demonstrated by the predictive checks plots, with the sum and standard deviation of the observed total number of positive samples falling well within the predicted values both for the total dataset (Appendix A) and for the different categories in all variables considered.

For the response variable of testing positive for target serovars, we were unable to achieve model convergence. This could be partially because of the very low number of positive observations (i.e., sampling events in which target serovars were detected) in the database (0.4% of all sampling events). However, when the predictions generated by the model fitted with the detection of *Salmonella* (any serovar) as the outcome of interest were used to predict the positivity to target serovars, good sensitivity and specificity were still found as demonstrated by the ROC and precision–recall plots (Appendix A).

## 4. Discussion

The importance of poultry as a major source of foodborne salmonellosis has triggered in the past the implementation of control programs in multiple countries aiming at minimizing *Salmonella* infection in poultry production in order to prevent the contamination of meat and eggs [19]. In the EU, the application of these programs in laying hens, which currently aim at a reduction target of ≤2% positive flocks to serovars of public health importance, led to a steep reduction in flock prevalence of *Salmonella* spp. and target serovars between 2008 and 2014 (down to 2.1% and 0.9%, respectively), but since then, flock prevalence has increased slightly and stabilized around 3.3% (*Salmonella* spp.) and 1.2% (target serovars) [1]. In this context, and considering the ongoing changes experienced by the industry in the last decades (with past changes banning use of battery cages in favor of enriched cages and the current trend following the “End of cage age” initiative) [20], it is of paramount importance to identify factors associated with increased risk of *Salmonella* infection in poultry production, including the characteristics of the surveillance activities themselves (e.g., which samples are collected, by who, etc.) in order to optimize control programs. Here, we performed an exhaustive analysis of the data generated in the Spanish NCP to describe the characteristics of farms/flocks/sampling events positive for *Salmonella* spp. and for target serovars and also to identify factors associated with an increased probability of detection. 

In this study, several factors at the farm, flock and sampling level were strongly associated with the risk of detecting *Salmonella* spp. Results obtained for the only variable at the farm level, i.e., the **location of the farm** (autonomous region), suggested that farms located in certain regions could experience baseline odds of positivity up to four times larger than others (Table 2). This geographic variation, which can be also observed at the European level [1], could be related to farm-level characteristics not considered in the model (that is, different from flock size or housing type) but also to environmental conditions associated with certain regions such as more extreme temperatures, which could lead to increased stress and in turn increased risk of spread and persistence of *Salmonella* [21]. 

Among the flock-level covariables, **housing type** was strongly associated with an increased risk of detecting *Salmonella* spp., with flocks in cages experiencing a larger risk compared with other categories. Multiple observational studies to assess the effect of the housing system on the prevalence of *Salmonella* in laying hens have been conducted in the past with conflicting results, which could be due to differences in methodologies and sample sizes considered [22]. Similar results to those described here were found in a large EU baseline study in laying hens [23] and other smaller observational studies in several European countries [11,24,25,26], with hens raised in cages experiencing significantly higher odds of contamination with *Salmonella*, although these studies considered mostly laying hens in conventional (battery) cage systems, which were phased out in 2012. This increased risk in caged systems could be attributed to difficulties in effectively cleaning cages and a higher infectious pressure derived from higher animal densities leading to increased shedding, at least for some serovars [27,28,29]. In contrast, other studies have indicated no differences or even higher odds of *Salmonella* contamination in laying hens and/or eggs from hens housed in non-caged vs. caged systems [30,31,32], which could be due to an increased risk of *Salmonella* infection from environmental sources [33]. Our results demonstrate that, at the national level, currently the risk of *Salmonella* spp. detection in caged flocks is approximately twice as high (95% PPI 1.52–2.85) compared with non-caged systems; however, it was not possible to further discriminate between the risk among the different varieties included in the non-caged category (organic, free-range and barn housing systems). 

The proportion of *Salmonella* spp.—positive sampling events was higher for flocks > 10,000 birds (though not when considering target serovars, Table 1), in agreement with previous work indicating that larger flocks (>20,000 or 30,000 birds) were subjected to increased risk of *Salmonella* detection [25,34]. However, **flock size** was not included in the multivariable model due to a lack of convergence when both this variable and housing type were considered at the flock level likely due to their high correlation, and housing type was preferred. 

**Vaccination** of flocks with both *S*. Enteritidis and non-*S*. Enteritidis vaccines has been previously linked with a decreased risk of contamination of the farm compared with unvaccinated flocks [10]. In Spain, there are several *S*. Enteritidis and/or Typhimurium-based attenuated and inactivated vaccines that can be used in poultry, and in fact, vaccination is compulsory in pullets (rearing phase) of laying hens except in farms with adequate biosecurity and subjected to appropriate *Salmonella* controls (with negative results for target serovars) according to Commission Regulation 1177/2006. Unfortunately, no specific information regarding the vaccination status of the flocks included in the study was available, and therefore, the effect of this practice in our results could not be assessed here. Nevertheless, vaccination coverage in layer flocks in Spain is close to 100% (Ministerio de Agricultura, Pesca y Alimentación, personal communication), and therefore, it should not have had a major impact on the differential risks observed in certain cases. 

Regarding variables at the sampling level, the analysis of the **sampler effect** (FBOp or CA) confirmed what had been already described in several EU Member States for broiler and breeder and fattening turkey farms [1]: a substantially increased odds of obtaining a positive result when a periodical sampling was conducted by CAs compared with FBOps (OR = 6.64, 95% PPI 5.63–7.75) was found once other known risk factors were accounted for in the model. This difference was even larger when compared with FBOp environmental sampling events (Table 2), which is not unexpected since these represent a different epidemiological scenario (barns recently emptied and subjected to cleaning and disinfection). This reflected the difference observed in the proportion of *Salmonella* spp. sampling events depending on the sampler (almost five times higher for CAs than for FBOps considering periodical sampling events, Table 1), which was nevertheless lower than what was found in the *Salmonella* NCPs in Spain in 2020 for other poultry species (2.1% CA vs. 0.07 FBOp in broiler flocks and 3.9% CA vs. 0% in FBOp in fattening turkeys, while no positive breeder flocks were detected by either) [1]. This difference could be at least in part due to the usual practice of analyzing independently more than one sample in sampling events conducted by the CA (93.6% considering more than one independent analysis) compared with those conducted by the FBOp (0.4% and 5.6% for periodical and environmental sampling events, respectively) since testing more samples can help to maximize the probability of detecting infected holdings as demonstrated in the EU baseline study, in which only one or two positive samples were detected in 38% of around 1540 *Salmonella* spp. positive flocks [23]. Sampling events conducted by the CA were more often focused on older flocks (e.g., 35% of all CA periodical sampling events were conducted in flocks > 83 weeks compared with 18% in FBOp periodical sampling events), but this should not have influenced the estimates provided here since the age of the flock was also included in the model. The same laboratory procedures for *Salmonella* detection are used regardless of the sampler (CA or FBOp) and regardless of the laboratory where this process is conducted (with some laboratories processing both samples collected by CAs and FBOps), and thus, it is unlikely that the sensitivity of the culture method is affected by this. However, the sensitivity of sampling events conducted by FBOps could be influenced by the more heterogeneous expertise of the personnel that can be involved in sample collection in FBOp sampling events compared with those conducted by CAs, which could in part also explain our results. 

A strong effect of the **age** of the animals on the odds of positivity was also observed, with older (>83 weeks) hens having 1.63 (95% PPI 1.35–2.00) higher odds of testing positive for *Salmonella* spp. compared with birds < 39 weeks (Table 2). Previous studies have found similar results pointing at a higher risk of infection in older animals [24,35], which could be due to an increased bacterial load in the environment (accumulation of *Salmonella*), an extended opportunity to be exposed to the bacteria and/or increased susceptibility as animals age (coupled with a decrease in the protection conferred by vaccination administered as pullets), although this effect has been also linked with moulting (a practice not allowed in the EU) in some cases [36]. However, this effect was not found in some cases [7], highlighting the difficulty in comparing studies based on different populations (also regarding age ranges considered and ways of considering these in the analysis). 

Both **the year and the time of the year** in which the sampling was carried out were also associated with the results. The year effect was mostly due to the lower risk of positive results for *Salmonella* spp. in sampling events performed in the first year of the study period (2015) compared with the reference (2020), while the odds of positivity were very similar for the rest of the study period (Table 2). In the case of season, however, a strong effect was observed for the latter months of each year (November–December and particularly September–October). A seasonal effect on the probability of foodborne salmonellosis and *Salmonella* infection in layer farms has been described in the past [6,37,38], though typically linked to summer months, which could be attributed in part to the thermal stress caused by higher temperatures that would lead to increased shedding, even though this would only apply to flocks in which the temperature inside the barn cannot be perfectly regulated (e.g., systems providing outdoor access). Nevertheless, this seasonal effect has been also connected to other causes linked with specific periods of the year that would affect the management or distribution of living vectors such as rodents or insects [7,8]. Indeed, sampling in winter was associated with increased detection of *Salmonella* in a study including five European countries, which was attributed to lower air quality in winter and a higher proportion of animals staying inside (for systems with outdoor sections) due to adverse climatic conditions [26]. Further studies are needed to better understand which changes could lead to this increased risk of detection in fall/early winter in Spain, which were also observed when comparing the raw proportion of positive sampling events for target serovars (Table 1). 

The **type of sample** analyzed has been also previously found to influence *Salmonella* recovery rates in the laboratory, with dust yielding a higher proportion of positive samples compared with fecal samples [23,24,34], which could be due to the increased survivability of *Salmonella* in dust compared with other *Enterobacteriaceae* [12]. This is in contrast with our results, since dust (and fabric swab)-based sampling events had slightly lower odds of retrieving *Salmonella* spp. while boot swabs and fecal samples had a more similar performance (Table 2, Figure 3). This result is, however, not unexpected considering that 97.8% of the dust and fabric swab samples were collected in the frame of environmental sampling events conducted by FBOps, which yielded the lower proportion of positive sampling events for *Salmonella* spp. (Table 2). In any case, this result should be interpreted carefully since dust and fabric swabs represented only ~12% of the sampling-level dataset, and furthermore, this information was already partially accounted for in the sampler/reason for sampling (since almost all of them were identified as FBOp environmental sampling events, mostly performed for the verification of the cleaning and disinfection process). Moreover, an effect of the housing system in the performance of dust-based sampling has been described, with lower sensitivity in sampling events based on dust samples compared with feces in non-cage flocks [39], further highlighting the complex interaction between several covariables in the probability of *Salmonella* detection. 

Only a small proportion (11.0%) of all positive sampling events detected in 2015–2020 was due to the presence of target serovars, in agreement with previous data from NCPs on breeding flocks in which target serovars (also including Hadar, Infantis and Virchow in this case) were detected in around 10% of all positive flocks in Spain in 2014–2016 [6]. Nevertheless, several of the non-target serovars retrieved (such as Infantis, Agona, Kentucky and Newport, found in 136, 29, 20 and 18 sampling events) were among the top 20 most frequent serovars in confirmed human cases described in Europe in 2020 [1], and in fact, serovars other than Enteritidis and Typhimurium/monophasic have been previously associated with foodborne *Salmonella* outbreaks linked with eggs, albeit at a much lower frequency compared with Enteritidis [40]. This, linked with the ability of the model fitted using *Salmonella* spp. detection as the outcome variable to predict the risk of detection of target serovars, suggests that results found here may have public health significance. Nevertheless, there may not be a clear linkage between the detection of several serovars that may enter the flock through very diverse routes such as environmental contamination and human cases, and therefore, the significance of our findings must be interpreted carefully. Still, multiple previous studies aiming at understanding the epidemiology of *Salmonella* in laying hens have also considered other serovars as the target variable often leading to the identification of shared risk factors with those increasing the risk of infection by target serovars [10], further supporting the approach followed here to identify variables to be considered for the optimization of surveillance and monitoring activities. 

Several limitations must be considered in the interpretation of the results obtained in this study: two-way interactions were not included in the model to avoid increasing its complexity derived from the hierarchical structure of the dataset. In addition, information on the specific farm locations was not available for the analysis, and thus, spatial autocorrelation could not be formally considered in the analyses. Positive spatial autocorrelation in the probability of detection of *Salmonella* in broiler farms located within 20 km was previously demonstrated in a study in Brazil focusing on a single vertically integrated company located in a specific region of the country [41], and in fact, the strong association of the region with *Salmonella* detection found in our study points at the possible existence of other spatially structured risk factors that were not included in the analysis and that should be further investigated. Temporal autocorrelation in the detection probability was not formally considered in the model either, though the lack of independence between the observations was accounted for through the use of random effects at different levels in the model. These farm- and flock-level random effects contributed to explaining the sampling-level risk, further suggesting that factors other than those explicitly considered here were also associated with an increased probability of positivity. Other studies have demonstrated that a previous *Salmonella* infection was linked to an increased risk of detection of *Salmonella* [7,34], which could help explain farm (and flock)-specific higher baseline risks. In any case, the inclusion of random effects to account for the lack of independence between observations in models evaluating the probability of *Salmonella* detection in poultry has been widely used before [34,42,43,44], supporting the validity of the approach followed here, which was intended to allow for a global analysis of the country-level data.

## 5. Conclusions

This study represents the first comprehensive description and analysis of a multi-year database including all laying hen flocks in Spain subjected to routine sampling in the frame of the *Salmonella* NCPs.

Our results demonstrate that, even though *Salmonella* detection was rare (especially involving the target serovars Enteritidis and Typhimurium/monophasic), there were certain farm-level (location), flock-level (housing system) and sampling-level variables (age of animals, sample type, sampler entity, year and bimester) that had a strong influence on the probability of detecting *Salmonella* (with an increased risk in flocks housed in caged flocks with older animals and in sampling events conducted by the CA in the last months of the year). Some of these factors had been previously identified in other countries/regions of the world, sometimes considering different production systems. Therefore, our findings can serve as a first step in the identification of factors associated with a differential probability of detection of *Salmonella* in the frame of the NCP in Spain that can be useful both to maximize the sensitivity of sampling strategies and to implement management measures destined to minimize the risk of infection on flocks presenting certain risk factors. 

## Figures and Tables

**Figure 1 animals-13-03181-f001:**
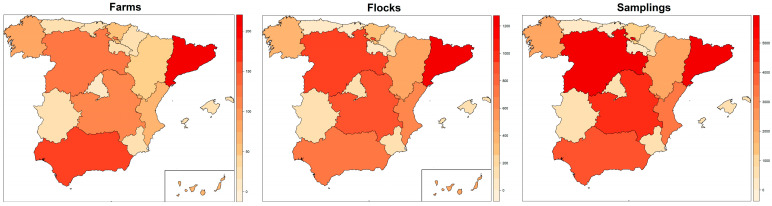
Number of farms, flocks and samplings analyzed in the frame of the national control program for Salmonella in laying hens between 2015 and 2020 in Spain per autonomous region.

**Figure 2 animals-13-03181-f002:**
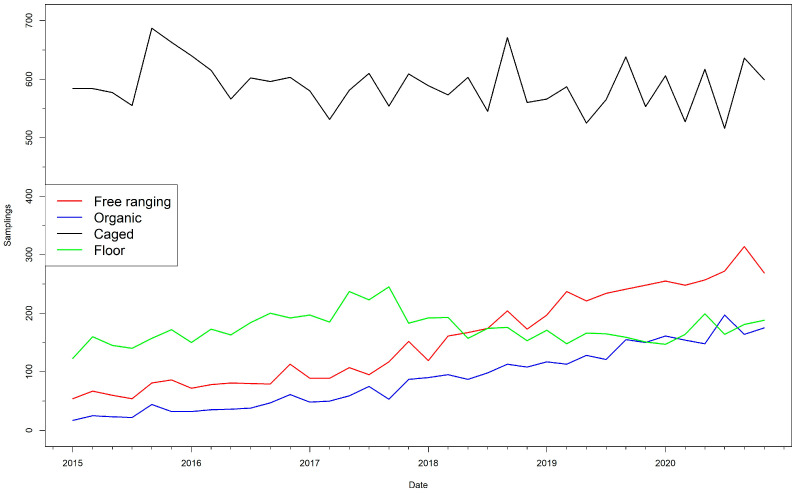
Number of sampling events per bimester conducted in flocks in the frame of the national control program for *Salmonella* in Spain between 2015 and 2020 depending on the housing system.

**Figure 3 animals-13-03181-f003:**
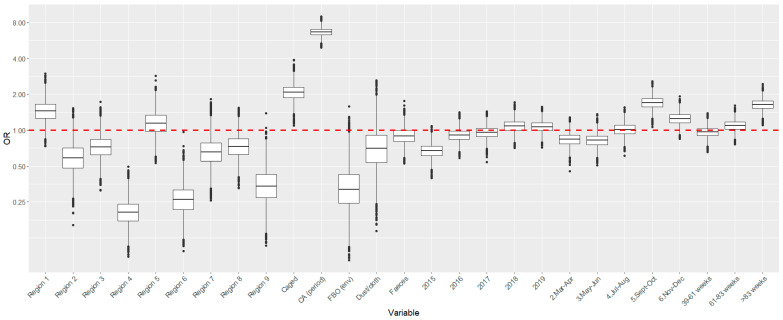
Estimates of the exponentiated coefficients (ORs) of the variables included in the final multivariable model using detection of *Salmonella* (any serovar) as the outcome and fitted with the data from *Salmonella* national control programs in Spain between 2015 and 2020 (*Y*-axis in the logarithmic scale).

**Table 1 animals-13-03181-t001:** Characteristics of farms, flocks and sampling events included in the *Salmonella* national control program in Spain 2015–2020 and proportion of those testing positive for *Salmonella* (any serovar and target serovars—Enteritidis, Typhimurium and its monophasic variant).

Level	Variable	Category	N (%)	*Salmonella* spp. Positive (%)	Target Serovar Positive (%)
Farm (n = 1153)	Region	Region 0	206 (17.9)	84 (40.8)	16 (7.8)
		Region 1	173 (15.0)	86 (49.7)	28 (16.2)
		Region 2	53 (4.6)	24 (45.3)	6 (11.3)
		Region 3	112 (9.7)	54 (48.2)	7 (6.3)
		Region 4	124 (10.8)	28 (22.6)	4 (3.2)
		Region 5	75 (6.5)	45 (60.0)	14 (18.7)
		Region 6	97 (8.4)	14 (14.4)	3 (3.1)
		Region 7	84 (7.3)	33 (39.3)	12 (9.1)
		Region 8	164 (14.2)	56 (34.1)	15 (9.1)
		Region 9	65 (5.6)	13 (20.0)	3 (4.6)
Flock (n = 7216)	Size	Q1 (<2500)	1810 (25.1)	170 (9.4)	36 (2.0)
		Q2 (2500–10,000)	1817 (25.2)	172 (9.5)	35 (1.9)
		Q3 (10,000–32,055)	1784 (24.7)	291 (16.3)	36 (2.0)
		Q4 (>32,055)	1805 (25.0)	205 (11.4)	16 (0.9)
	Housing type	Caged	3833 (53.1)	586 (15.3)	62 (1.6)
		Floor	1465 (20.3)	86 (5.9)	19 (1.3)
		Free-ranging	1188 (16.5)	86 (7.2)	29 (2.4)
		Organic	730 (10.1)	80 (11.0)	13 (1.8)
Sampling (n = 36,193)	Sampler/reason sampling	FBOp—periodical	28,481 (78.7)	730 (2.6)	23 (0.08)
		FBOp—env	4313 (11.9)	45 (1.0)	6 (0.14)
		CA—periodical	3399 (9.4)	430 (12.7)	103 (3.0)
	Year	2015	5112 (14.1)	148 (2.9)	8 (0.16)
		2016	5436 (15.0)	198 (3.6)	23 (0.42)
		2017	5756 (15.9)	199 (3.5)	21 (0.36)
		2018	6175 (17.1)	215 (3.5)	26 (0.42)
		2019	6556 (18.1)	224 (3.4)	29 (0.44)
		2020	7158 (19.8)	221 (3.1)	25 (0.35)
	Bimester	Jan–Feb	5796 (16.0)	148 (2.6)	11 (0.19)
		Mar–Apr	5792 (16.0)	151 (2.6)	13 (0.22)
		May–Jun	5910 (16.3)	159 (2.7)	18 (0.30)
		Jul–Aug	5903 (16.3)	170 (2.9)	19 (0.32)
		Sep–Oct	6512 (18.0)	305 (4.7)	32 (0.49)
		Nov–Dec	6280 (17.4)	272 (4.3)	39 (0.62)
	Age (weeks)	Q1 (<39)	8912 (24.6)	216 (2.4)	21 (0.24)
		Q2 (39–61)	8947 (24.7)	238 (2.7)	16 (0.18)
		Q3 (61–83)	9223 (25.5)	309 (3.4)	31 (0.34)
		Q4 (>83)	9111 (25.2)	442 (4.9)	64 (0.70)
	Number of analyses	1	32,645 (90.2)	801 (2.5)	40 (0.12)
		2–10	3548 (9.8)	404 (11.4)	92 (2.6)
	Sample type	Feces	20,065 (55.4)	898 (4.5)	64 (0.32)
		Boot swabs	11,720 (32.4)	249 (2.1)	59 (0.50)
		Dust	3542 (9.8)	48 (1.4)	7 (0.20)
		Fabric swab	866 (2.4)	10 (1.2)	2 (0.23)

**Table 2 animals-13-03181-t002:** Estimates of the farm, flock and sampling-level covariates on the probability of obtaining a positive result for *Salmonella* in 36,193 sampling events conducted in the frame of the national control program in Spain in 2015–2020.

Level (n)	Variable	Category	Median β, γ, δ	Rhat	Median OR (95% PPI)
Farm (n = 1153)	Region	Region 0	Ref		
		Region 1	0.369	1.001	1.45 (0.97–2.13)
		Region 2	−0.539	1.001	0.58 (0.33–1.03)
		Region 3	−0.328	1.006	0.72 (0.45–1.1)
		Region 4	−1.590	1.004	0.2 (0.13–0.33)
		Region 5	0.132	1.001	1.14 (0.71–1.81)
		Region 6	−1.340	1.002	0.26 (0.15–0.46)
		Region 7	−0.419	1.003	0.66 (0.38–1.14)
		Region 8	−0.317	1.003	0.73 (0.48–1.11)
		Region 9	−1.085	1.006	0.34 (0.18–0.63)
Flock (n = 7216)	Housing type	Others	Ref		
		Caged	0.729	1.005	2.07 (1.52–2.85)
Sampling (n = 36,193)	Sampler/reason sampling	FBOp—periodical	Ref		
		FBOp—env	−1.142	1.001	0.32 (0.15–0.74)
		CA—periodical	1.893	1.001	6.64 (5.63–7.75)
	Year	2015	−0.400	1.005	0.67 (0.52–0.87)
		2016	−0.097	1.003	0.91 (0.71–1.15)
		2017	−0.045	1.002	0.96 (0.75–1.21)
		2018	0.081	1.002	1.08 (0.87–1.36)
		2019	0.066	1.001	1.07 (0.86–1.33)
		2020	Ref		
	Bimester	Jan–Feb	Ref		
		Mar–Apr	−0.180	1.001	0.84 (0.65–1.07)
		May–Jun	−0.195	1.002	0.82 (0.64–1.06)
		Jul–Aug	0.010	1.001	1.01 (0.79–1.29)
		Sep–Oct	0.531	1.002	1.70 (1.36–2.12)
		Nov–Dec	0.218	1.001	1.24 (0.99–1.56)
	Age (weeks)	Q1 (<39)	Ref		
		Q2 (39–61)	−0.039	1.002	0.96 (0.78–1.18)
		Q3 (61–83)	0.090	1.001	1.09 (0.90–1.33)
		Q4 (>83)	0.492	1.002	1.63 (1.35–2.00)
	Sample type	Boot swabs	Ref		
		Feces	−0.116	1.004	0.89 (0.65–1.23)
		Dust/fabric swab	−0.350	1.001	0.7 (0.31–1.51)

## Data Availability

Data presented in this study are available upon reasonable request from the authors.

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
