# Peer review of "Risk Factors for Salmonella Detection in Commercial Layer Flocks in Spain"

_animals, 2023, doi:10.3390/ani13203181_

Round 1
Reviewer 1 Report
Dear authors,
I have read your manuscript "Risk Factors for Salmonella detection..." with great interest. It describes a topic of current interest, potentially useful for all EU member states. I particularly like that the analysis is carefully developed, and the model math is clearly presented.
In the end I have only a handful of minor remarks, that I present below line by line:
L14: Monitoring itself does not prevent anything. It is a prerequisite for intervention strategies though.
L159: Why was flock size not considered as a numeric variable?
L200-209 It took me a while to realize that the two outcome variables were the outcomes for two different models, thinking that you would model them jointly. Please add a line like "Two models with different outcome variables were considered..."
214 Bernouilli -> Bernoulli. Also put all math which is not variables (Bernoulli, logit, farm, flock....) in text mode i.e. not italic
225 Please put _farm and _flock in proper subscript
237: I'm not enthousiastic about the POPR. A median odds bigger than one (risk-factor) gives a positive POPR. And a median odds below one (beneficial effect on prevalence) gives a negative POPR. But a high POPR of 80% will become a low one when you switch the labels of your categorical variables. See e.g. table 2, it suggests that there are 2 super risky regions, while in fact choosing another baseline region would give another number of high POPR regions. So it is a bit arbitrary. The median OR plus confidence interval gives all the information we need.
L248: the phrasing "final model" suggests that a variable selection was performed. Please describe this variable selection procedure. Do this before the implementation details (before line 242). How can POPR be used for variable selection, given that it depends on you choice of baseline level?
L255-257 leftover text from the template
L368 Here it is stated that variables may be excluded when correlated with other variables. This was not mentioned in the M&M (ony removal based on POPR and DIC). Was this correlation check performed before fitting? What method was used, and was the criterion for significant correlation?
Table 2: perhaps clearer to put 'reference'or 'baseline' instead of NA
Figure 2 caption: 2016 -> 2019
L442: I don't think you can speak of a four times higher risk. You get odds ratios which are not the same as risk ratios.
L520: about the effect of age: could it not simply be that the older an animal gets the opportunity for infection it had?
L540 hotter temperatures -> higher temperatures. Additionally: I'm not familiar with the housing system in Spain, but in our country we lock the animals up so thoroughly that they will never experience any outside temperature. Summer or winter, the hen will always experience the climate as set by the farm management.
L542: with 'living vectors' you mean flies, rodents, etc?
L617 in -> on
Reviewer 2 Report
The article titled “Risk factors for Salmonella detection in commercial layer flocks in Spain” provides useful information for the poultry industry.
The subject of the study is interesting and falls within the scope of the magazine.
Sections: "simple summary" is complete. "Abstracts" results are described too generally.
The "introduction" part is a good introduction to the whole topics.
The aim of the work formulated in finish of the introduction, , is not objectionable.
The "materials and methods" section is clear and detailed.
Data analysis and interpretation of results appear to be appropriate and accurately interpreted.
The 'results and discussion' section contains the scientific results, presented in two tables and two figures.
The results – though they are mostly neutral – are scientifically interesting.
The references section is all referenced in the manuscript.
The study adds to existing knowledge on the identification of factors associated with a differential probability of detection of Salmonella.
